# Innovative Fluorinated Polyimides with Superior Thermal, Mechanical, and Dielectric Properties for Advanced Soft Electronics

**DOI:** 10.3390/polym17030339

**Published:** 2025-01-26

**Authors:** Yuwei Chen, Yidong Liu, Yonggang Min

**Affiliations:** 1School of Electromechanical Engineering, Guangdong University of Technology, Guangzhou HEMC, No. 100 Waihuanxi Road, Guangzhou 510006, China; polebear1996@outlook.com; 2Widerange Flight Engineering Science and Applications Center, Institute of Mechanics, Chinese Academy of Sciences, No. 15 Beisihuanxi Road, Beijing 100190, China; 3Guangdong Aerospace Research Academy, Nan Sha, Guangzhou 511458, China

**Keywords:** fluorinated, polyimide, low dielectric constant, molecular dynamics, DFT

## Abstract

This study addresses the limitations of traditional polyimides (PIs) in high-frequency and high-temperature soft electronic applications, and then introducing trifluoromethylbenzene (TFMB) into the molecular structure and employing various diamines as connecting components to solve the bottleneck. The innovative molecular design enhances thermal, mechanical, and dielectric properties, overcoming challenges in balancing these performances. The optimized fluorinated PI (TPPI50) exhibits exceptional properties, including a glass transition temperature of 402 °C, thermal decomposition temperature of 563 °C, tensile strength of 232.73 MPa, elongation at break of 26.26%, and dielectric constant of 2.312 at 1 MHz with a dielectric loss as low as 0.00676. These improvements are attributed to the unique synergy between TFMB’s fluorinated groups, which reduce molecular polarization, and the biphenyl structure, which reinforces chain stability. Compared to conventional PIs, TPPI50 demonstrates superior comprehensive performance, making it highly suitable for soft circuits, high-frequency signal transmission, and advanced applications such as wearable devices and biosensors. This study provides a robust framework for industrial applications, offering a path to next-generation soft electronics with enhanced reliability and performance.

## 1. Introduction

Soft electronics are devices engineered to operate reliably under deformation conditions such as bending, stretching, or compression. In contrast to conventional rigid electronic devices, soft electronics are distinguished by their stretchability. Their primary advantages include conformability to complex three-dimensional shapes, as well as being thin, durable, and comfort [1,2]. The advancement of soft electronics is underpinned by innovations in soft substrate materials, conductive materials, and device architectures. With continuous progress in materials science, nanotechnology, and electronic engineering, for example, the application of folding screens in mobile phones, the use of nanomaterials in flexible electronics, and portable devices with a high folding ratio, soft electronics are increasingly overcoming the constraints of traditional technologies, driving breakthroughs in areas such as smart wearables, health monitoring, intelligent medical systems, and human–computer interactions [3,4,5].

Polyimides (PIs) have emerged as key materials in the field of soft electronics due to their outstanding thermal stability, electrical insulation, and mechanical strength [6]. As a high-performance engineering plastic, PIs exhibit exceptional resistance to high temperatures, radiation, and chemical degradation while maintaining excellent flexibility, making them ideal for soft electronic applications [7,8]. In these devices, PIs serve as base materials capable of withstanding deformation, such as bending and stretching, without compromising their electrical insulation properties. Their superior heat resistance ensures stable electrical and mechanical performance even at elevated temperatures, rendering them particularly suitable for soft electronic devices requiring long-term reliability. However, as soft electronics continue to evolve, the inherent dielectric properties of PIs are increasingly insufficient to meet the demands of high-frequency signal transmission. Functional modifications, such as the incorporation of nanofillers or the introduction of fluorinated groups, are necessary to address these limitations [9,10,11,12].

The dielectric constants of PIs can be significantly reduced through the introduction of fluorinated groups, primarily due to alterations in their molecular structure and electronic properties. First, the high electronegativity of fluorinated groups (e.g., -CF, -C6F6) markedly decreases the polarization of molecular chains [13]. The electron cloud confinement in C-F bonds weakens their polarization ability, thereby lowering the dielectric constant. Second, the bulky nature of fluorinated groups increases the spacing between molecular chains, reducing intermolecular dipole interactions and further diminishing charge transfer effects [14,15,16]. Additionally, the highly symmetrical structures introduced by certain fluorinated groups (e.g., -C6F6) minimize contributions from molecular orientation polarization, effectively lowering the overall dipole moment. Collectively, these mechanisms enable fluorinated groups to achieve a substantial reduction in the dielectric constant of PIs at the molecular level by decreasing molecular polarity, limiting electron polarization, and mitigating dipole interactions [17,18].

In terms of mechanical properties, the rigidity of the C-F bond enhances the stiffness of the polymer chain by restricting intramolecular rotations and motions. This increased chain rigidity translates into higher elastic modulus and tensile strength, improving the material’s ability to withstand mechanical deformation [19]. However, the large atomic size of fluorine and the associated increase in free volume can sometimes reduce intermolecular van der Waals forces, potentially lowering the material’s elongation at break and flexibility [20,21]. Thus, while fluorinated groups reinforce the polymer mechanically, there is a trade-off in terms of ductility that must be carefully balanced through molecular design.

The thermal properties of PIs are also significantly influenced by the presence of C-F bonds. The high bond dissociation energy of the C-F bond (~485 kJ/mol) compared to C-H and C-C bonds imparts superior thermal stability to the polymer, increasing both the glass transition temperature (Tg) and thermal decomposition temperature (T5%) [22,23]. These characteristics enable fluorinated PIs to maintain structural integrity and functionality under high-temperature conditions, making them highly suitable for applications in harsh environments [24,25,26].

In addition, fluorinated groups also enhance the hydrophobicity of PIs. The low surface energy of C-F bonds reduces water absorption, which not only improves the chemical resistance of the polymer but also protects it from moisture-induced degradation [27,28]. This is particularly advantageous for applications in humid or corrosive environments, where long-term stability is required. Moreover, the presence of fluorinated groups on the polymer surface can contribute to anti-fouling properties, further broadening the scope of application [29,30].

Fluorinated PIs also exhibit excellent resistance to radiation due to the strong bond energy and chemical inertness of the C-F bond [31]. Radiation-induced bond breakage is significantly minimized, enabling the polymer to retain its mechanical and dielectric properties in high-energy environments such as outer space or nuclear facilities. This resistance makes fluorinated PIs highly attractive for aerospace, defense, and advanced electronic applications.

Our experimental investigations [32] have systematically evaluated the effects of different fluorinated groups on the properties of PIs. While these modifications successfully reduced the dielectric constant to below 2.3, the thermal and mechanical properties did not meet expectations. To achieve a balance between a low dielectric constant and a superior thermal and mechanical performance, this study introduced various diamines as connecting components (CCs) and incorporated TFMB into the PI molecular chain. Through molecular structure redesign, we conducted in-depth research employing molecular dynamics simulations, density functional theory (DFT) calculations, and experimental characterizations. The performance of fluorinated PI systems with varying TFMB content and molecular structures was systematically verified.

The results indicate that, when the CC is p-PDA, the TFMB-containing PI (TPPI) system exhibits the best overall performance. Notably, TPPI50, with a TFMB content of 50%, stands out with exceptional properties: a glass transition temperature (Tg) of 402 °C, a thermal decomposition temperature (T5%) of 563 °C, a tensile strength of 232.73 MPa, an elongation at break of 26.26%, and an elastic modulus of 5.53 GPa. Furthermore, TPPI50 demonstrates excellent electrical performance, with a dielectric constant (Dk) of only 2.312 and a dielectric loss (Df) as low as 0.00676. These values are significantly superior to those of traditional PI materials. The exceptional combination of thermal, mechanical, and electrical properties positions TPPI50 as a promising material for applications in soft circuits and high-frequency signal transmission. This advancement not only enhances the performance and reliability of soft electronic devices but also lays a solid foundation for innovative applications in smart wearables, electronic skin, flexible displays, and biosensors.

## 2. Materials and Methods

### 2.1. Simulation

#### 2.1.1. Model Establishment

The structures of dianhydride and diamine monomers were constructed using the Visualizer module of BIOVIA Materials Studio (MS) 2019. These monomers were then utilized to create various repeating units via the Forcite Geometry Optimization code within MS 2019. The optimized structures are presented in Table 1. The COMPASS II force field was employed throughout the modeling and simulation process.

The initial density of the model box was set to 0.1 g/cm^3^. In the first step, the NPT ensemble was applied using the Berendsen method, with the pressure and temperature set to 0.5 GPa and 298 K, respectively, until the system density stabilized. In the second step, an annealing process was performed within a temperature range of 300 K to 500 K. In the third step, the NPT ensemble was re-applied using the Berendsen method with the pressure and temperature set to 0.0001 GPa and 298 K, respectively, until the system reached equilibrium. All modified PI structures discussed in this paper are illustrated in Figure 1.

#### 2.1.2. Mechanical Property Simulation

The mechanical properties of the materials were simulated using the Mechanical Property option within the Forcite module. The constant strain method was employed to calculate the elastic modulus, tensile strength, and other mechanical properties along various directions. This approach estimates the elastic constant matrix (stiffness matrix) through a series of finite difference approximations.

The process began by removing symmetry from the system, followed by the optional re-optimization of the structure, during which cell parameters could be adjusted (as shown in Figure 1). In this method, each strain pattern is represented in Voigt notation and subsequently converted into a strain matrix (ε). The metric tensor (G) was then generated using the following formula:(1)$$G=H_{0}^{\prime }\left[2\varepsilon +1\right]H_{0}$$

Here, *H*_0_ is the original lattice vector matrix, I is the identity matrix, and $H_{0}^{\prime }$ is the transpose of *H*_0_. The new lattice parameters derived from G contributed to the establishment of a stiffness matrix, which relates the applied strain to the resulting stress patterns.

#### 2.1.3. Polarizability

The molecular structures established in MS 2019 were imported into Gaussian View and optimized using the PBE1PBE [33] functional with a 6-311G** [34,35] basis set. These optimized structures were subsequently used for further polarizability calculations employing the PBE1PBE functional with the aug-cc-pVTZ [36,37] basis set. Relevant parameters were then extracted, and polarizability values were calculated using the Multiwfn software (3.8 dev WIN64) [38,39].

### 2.2. Experiment

#### 2.2.1. Chemicals and Materials

3,3′,4,4′-Biphenyltetracarboxylic dianhydride (BPDA, 99.9%) and p-Phenylenediamine (p-PDA, 99.5%) were purchased from CHINATECH (Tianjin, China) Chemical Co., Ltd. 9,9-Bis(4-aminophenyl)fluorene (BAFL, 99%) was obtained from Aladdin, Shanghai, China. Other chemicals, including 4,4′-Oxydianiline (ODA, 99%), 4,4′-(9-Fluorenylidene)dianiline (FDA, 99%), 4-Phenylethynyl phthalic anhydride (4-PEPA, 98%), 1,3-Bis(3-aminophenoxy)benzene (APB, 98%), 4,4′-Oxydiphthalic anhydride (ODPA, 97%), and N,N-Dimethylacetamide (DMAc, analytically pure), were procured from Macklin, Shanghai, China.

#### 2.2.2. Preparation of TPIn

TFMB and BPDA were polymerized with various connecting components (CCs), such as p-PDA and ODA, in a DMAc solution. Two kinds of CCs, ODA and PDA, are selected in this paper, because this paper mainly studies the effects of the C-F bond and CF_3_ side group of TFMB on molecular structure and PI performance. Both ODA and PDA are symmetrical structures and have no side groups, which do not interfere with other structural factors for the main research purpose. At the same time, ODA has a flexible structure, while PDA has a rigid structure, which is two representative CC structures. Under the same TFMB content, the difference between rigid CC and flexible CC in PI performance can be compared.

The specific experimental process is as follows. The molar ratio of each CC to TFMB is set to 1/3, 1/1, and 3/1. TFMB and CCs were stirred at a certain rate in the DMAc solution at room temperature in the proportions described above until they completely dissolved. Then, the TPAA solution was prepared by slowly adding BPDA, and the reaction lasted for 8 h.

Subsequently, the TPAA solution was coated on a high borosilicate glass plate and treated at 100 °C, 200 °C, and 300 °C for 1 h in order to prepare TPIn thin films. The films were named TOPI25/TOPI50/TOPI75 and TPPI25/TPPI50/TPPI75 according to the type of CC and the molar ratio of TFMB to CC. All the thicknesses of the films after 100 °C were controlled within the range of 110–125 μm using a coating rod of a specific thickness; thus, the final thickness of TPIn films were maintained at 20–25 μm.

### 2.3. Characterization and Measurement

The chemical structures of TPIn were analyzed using Fourier-Transform Infrared Spectroscopy (FTIR) with a Nicolet 6700 FTIR Spectrometer (Thermo Fisher Scientific, Waltham, MA, USA), equipped with an Attenuated Total Reflection (ATR) module, covering a spectral range of 4000 to 400 cm^−1^. Thermogravimetric Analysis (TGA) was performed on a TGA/DSC 3+/1600HT instrument (Mettler-Toledo International Inc., Columbus, OH, USA) at a heating rate of 10 °C/min, from 30 °C to 750 °C, under a nitrogen atmosphere. Differential Scanning Calorimetry (DSC) was conducted using a DSC3 instrument (Mettler-Toledo International Inc., Columbus, OH, USA) at a heating rate of 20 °C/min, from 30 °C to 450 °C, under a nitrogen atmosphere. The mechanical properties of the materials were measured using an AG-X Plus electronic universal testing machine (Shimadzu Corporation, Kyoto, Japan).

## 3. Results and Discussion

### 3.1. Structure

As shown in Figure 2a, the peaks at 1781 cm^−1^ and 1718 cm^−1^ correspond to the asymmetric and symmetric stretching vibrations of the C=O bond, respectively. The stretching vibration peak of the C-N bond, observed at 1363 cm^−1^, confirms the successful formation of the imide structure in the polymer. Additionally, the absorption peaks of the C-F bond at 1124 cm^−1^, 1176 cm^−1^, and 1240 cm^−1^ indicate that the -CF₃ structure has been successfully incorporated into the PI molecular framework. The full-band spectrum of FTIR (shown in Appendix A) indicates that except for the characteristic peaks mentioned above, there are no characteristic peaks of other impurity groups in 2000 cm^−1^~4000 cm^−1^ and 0~600 cm^−1^.

### 3.2. Thermal Properties

Figure 2b shows the thermogravimetric curves of TPIn, with detailed data summarized in Table 2. Increasing the TFMB content significantly enhances the Tg, 1% weight loss temperature (T_1%_), and 5% weight loss temperature (T_5%_). As illustrated in Figure 1c, TFMB introduces an additional benzene ring and six C-F bonds compared to p-PDA. The benzene ring, known for its high bond energy, provides excellent thermal stability, while the biphenyl structure further strengthens the molecular chain.

The C-F bond, with its higher bond energy compared to other common bonds (e.g., C-H and C-C), contributes to maintaining molecular integrity at elevated temperatures. As the TFMB content increases, the proportion of biphenyl structures and C-F bonds also rises, enhancing the stability of the TPPI molecular framework. This is reflected in a marked increase in decomposition temperatures, including T_5%_, and T_g_, enabling TPPI to maintain structural integrity under high temperatures. Experimental data reveal that TPPI75 exhibits a T_g_ of 407 °C and a T_5%_ of 570 °C.

In comparison, while TFMB contributes similarly to the thermal stability of TOPI, the lack of a biphenyl structure limits the improvement. For instance, the T_g_ of TOPI75 only increases to 375 °C. However, as shown in Table 2, TOPI exhibits higher T_1%_ and T_5_% values overall. Specifically, TOPI50 achieves a T_1_% of 517 °C, which may result from structural differences between the CC monomers. The additional benzene ring in ODA provides strong heat absorption, allowing TOPI to exhibit superior thermal performance at high temperatures compared to TPPI.

### 3.3. Mechanical Properties

The stress–strain curves of TPIn are shown in Figure 2c, with the corresponding mechanical characterization and simulation data provided in Table 2 and Table 3. When the connecting component (CC) is p-PDA, the TPPI molecular structure contains a higher proportion of benzene rings, which significantly enhances molecular chain rigidity and improves the material’s resistance to external forces. Additionally, the fluorinated C-F groups in TFMB, located on the benzene ring, further increase the spatial rigidity of the molecular chain, restricting its rotational and translational freedom. Consequently, TPPI50 demonstrates superior tensile strength and elastic modulus compared to TPPI25.

However, the large volume and strong electronegativity of fluorinated groups increase the molecular chain spacing, weaken van der Waals forces, and result in decreased tensile strength and elastic modulus for a higher TFMB content, while elongation at break increases. Under different TFMB content, the balance between the strengthening and weakening effects of fluorinated groups varies. Experimental data reveal that, at 50% TFMB content, the strengthening effect dominates, giving TPPI50 a tensile strength of 232.73 MPa, significantly higher than TPPI25’s 205.92 MPa. However, when the TFMB content increases to 75%, the weakening effect prevails, reducing TPPI75’s tensile strength to 167.71 MPa. These excellent properties help flexible electronic materials maintain structural integrity and stability when subjected to external forces. In the manufacture of flexible electronic devices, the mechanical properties of materials are crucial to the reliability and durability of the devices.

Notably, TPPI50 achieves the highest elongation at break (26.26%), while TPPI25 exhibits the highest elastic modulus (5.76 GPa). The improved elongation at break of TPPI50 may be attributed to its higher degree of polymerization, which enhances macroscopic mechanical properties. Conversely, the high elastic modulus of TPPI25 likely results from the dominance of the fluorinated groups’ weakening effect on the modulus.

For TOPI, the mechanical property trends align with those of TPPI. However, as the TFMB content increases, the weakening effect diminishes more gradually. This slower decline may result from the compensatory strengthening effect of fluorinated groups, which offsets the adverse impacts of reduced phenoxy flexible groups from ODA and the weakening effects of fluorinated groups.

The data in Table 4 adequately reflect the difference in the influence of ODA and PDA on the mechanical properties of TPIn films. As mentioned earlier, PDA is a CC with a rigid structure, and it can be seen that TPPI is significantly better than TOPI on various moduli in different directions, because the structural rigidity of TPPI is significantly stronger than TOPI. The Poisson ratio of TOPI is significantly higher than that of TPPI, which means that TOPI is more prone to transverse deformation within the range of elastic deformation, which corresponds to the flexible structure of ODA.

### 3.4. Dielectric Properties

Figure 2d,e depict the variations in dielectric constant (Dk) and dielectric loss (Df) of TPIn with frequency. Both Dk and Df decrease as frequency increases, governed by several mechanisms. Firstly, the dielectric constant and loss are determined by the material’s ability to polarize under an electric field. Orientation polarization, which involves the reorientation of dipole molecules in response to the electric field, has a slow response time. At higher frequencies, the rigidity and strong electronegativity of fluorinated groups limit molecular chain mobility, reducing the dipole moment’s ability to align with the field. This significantly diminishes the contribution of orientation polarization, leading to reduced dielectric constant and loss. Secondly, fluorinated groups exhibit extremely low polarizability, with highly stable electron cloud distributions and minimal response to external electric field disturbances. Under high-frequency fields, electron polarization becomes the dominant form of polarization. The inherently low electron polarization of fluorinated groups further lowers both Dk and Df. Thirdly, the volumetric effects of fluorinated groups increase the spacing between molecular chains, thereby reducing dipole–dipole interactions. As frequency rises, polarization coupling between molecules weakens further, minimizing dielectric loss.

At lower frequencies, energy loss primarily originates from orientation and interfacial polarization. The rigidity and electronegativity of fluorinated groups suppress these processes, reducing the loss. At higher frequencies, dielectric loss is dominated by conductivity loss and electron polarization. The fluorinated groups inhibit high-frequency conductivity loss by reducing charge migration and improving dielectric stability.

The Dk and Df values for each TPI film at 1 MHz are summarized in Table 2, with trends illustrated in Figure 2f. As the TFMB content increases, the Dk of TPI films at high frequency decreases significantly, with TPPI75 and TOPI75 reaching a Dk of just 2.22. However, Df no longer decreases notably and may even stabilize or slightly increase due to several factors.

Firstly, as the fluorinated group content increases, internal polarization mechanisms, such as orientation and interfacial polarization, reach saturation, rendering further improvements negligible. Secondly, the increased rigidity and volume of fluorinated groups enlarge molecular chain spacing, potentially promoting local charge accumulation or migration, which triggers new energy loss mechanisms. Additionally, excessive fluorinated groups may cause microscopic phase separation or defects within the material, forming weak conductive paths that increase conductivity loss, particularly under high-frequency or high-field conditions. Finally, a high concentration of fluorinated groups can disrupt molecular chain symmetry, complicate polarization behavior, and enhance surface and interfacial effects, further contributing to dielectric loss. Among all tested samples, TPPI50 and TOPI50 exhibited the lowest Df values (0.00676 and 0.00774, respectively), highlighting their superior balance of dielectric performance and structural integrity.

To validate the hypotheses mentioned earlier, the optimized models were imported into Gaussian for further structural optimization and polarizability calculations. The simulation results, obtained using the Multiwfn software, are shown in Table 5. It was observed that polarizability in polymers is anisotropic, with variations depending on the influence of different functional groups. TPPI exhibited a more pronounced degree of anisotropy compared to TOPI. Specifically, the polarizability of TPPI in the X and Z directions was greater than that of TOPI, while TOPI displayed significantly higher polarizability in the Y direction. This resulted in TOPI having a higher average volumetric polarizability than TPPI, indicating that, under the same frequency and TFMB content, TOPI’s Dk is larger than TPPI’s, consistent with the findings in Table 2.

Fluorinated groups exhibit extremely low polarizability. As the C-F bond content increases, the overall structural polarizability decreases further. Additionally, C-F bonds, typically located on benzene rings or side chains, reduce the molecular symmetry, indirectly lowering the structural polarizability. These observations confirm the earlier hypothesis regarding the correlation between changes in Dk and polarizability.

## 4. Conclusions

This study aimed to address the limitations of conventional polyimides (PIs) in meeting the advanced dielectric, thermal, and mechanical performance requirements of soft electronics, particularly in high-frequency and high-temperature applications. By introducing TFMB as a fluorinated component into the PI molecular structure and employing various diamines as connecting components (CCs), a series of fluorinated PIs were synthesized and systematically characterized to optimize their properties. Comprehensive characterization techniques, including FTIR, TGA, DSC, and mechanical and dielectric property testing, were employed to evaluate the structural, thermal, mechanical, and dielectric performance of the modified PIs. Molecular dynamics simulations and density functional theory (DFT) calculations were also conducted to provide theoretical insights into the observed trends.

The results revealed that the incorporation of TFMB significantly enhanced the thermal stability of the PI films, with TPPI75 achieving a Tg of 407 °C and a T5% of 570 °C due to the high bond energy of the C-F bonds and the biphenyl structure introduced by p-PDA. In terms of mechanical properties, TPPI50 demonstrated an optimal balance, with a tensile strength of 232.73 MPa, an elongation at break of 26.26%, and an elastic modulus of 5.53 GPa. The elongation at break of TPPI50 was notably higher, likely due to the enhanced polymerization degree and balanced molecular rigidity.

From a dielectric perspective, the introduction of fluorinated groups successfully reduced the Dk and Df. TPPI75 achieved a Dk as low as 2.22 at 1 MHz, while TPPI50 exhibited the lowest Df of 0.00676. These improvements were attributed to the reduced molecular chain polarization, enhanced rigidity, and decreased intermolecular dipole–dipole interactions caused by the fluorinated groups. However, the excessive TFMB content introduced some limitations, such as potential phase separation and minor conductivity loss at high frequencies, which were most pronounced in the 75% TFMB samples. Furthermore, the results of molecular dynamics and DFT simulation also provide strong structural theoretical support for the above inferences.

The innovation of this study lies in the precise molecular design strategy, combining fluorinated components and tailored CC monomers to simultaneously enhance thermal, mechanical, and dielectric properties. The optimized TPPI50 system demonstrates exceptional comprehensive performance, surpassing that of conventional PI materials (as shown in Table 6), and is particularly well suited for use in soft circuits, high-frequency signal transmission, and other advanced applications.

Looking ahead, the outstanding properties of TPPI50, including high thermal stability, mechanical durability, and superior dielectric performance, position it as a promising candidate for applications such as smart wearables, soft displays, electronic skins, and biosensors. The successful integration of TFMB into the PI molecular chain provides a robust framework for future research, enabling further optimization for specific application scenarios and expanding the scope of soft electronic technologies.

## Figures and Tables

**Figure 1 polymers-17-00339-f001:**
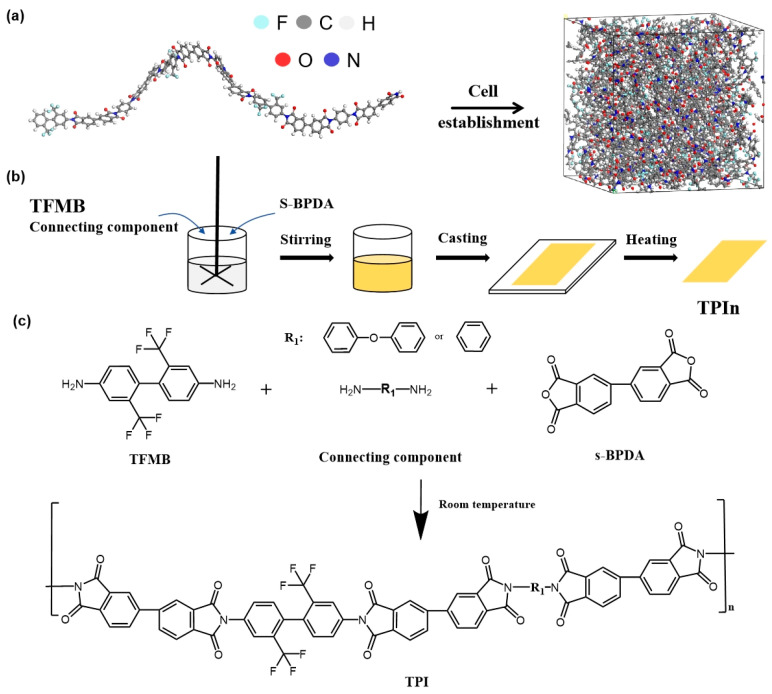
The diagrams of (**a**) TPI molecular structure, (**b**) experiment, and (**c**) synthesis route.

**Figure 2 polymers-17-00339-f002:**
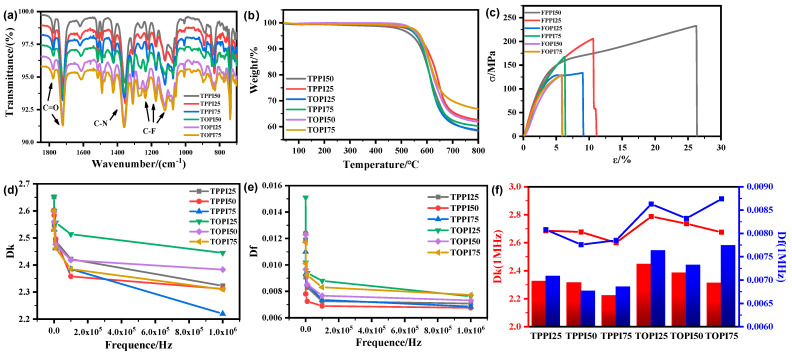
The TPIn spectras of (**a**) FT-IR; (**b**) TGA curves; (**c**) stress-strain curves; (**d**) Dk curves with frequency; (**e**) Df curves with frequency; and (**f**) the intensity of Dk and Df at 1 MHz.

**Table 1 polymers-17-00339-t001:** The specific setup of the simulation process.

Ensemble	Temperature	Timestep	Pressure	Thermostat	Barostat
NPT (compress)	298 K	1 fs	0.5 GPa	Nose–Hoover [33]	Berendsen [34]
Anneal	300 K~500 K	1 fs	-	-	-
NPT (balance)	298 K	1 fs	0.0001 GPa	Nose–Hoover	Berendsen

**Table 2 polymers-17-00339-t002:** Abbreviated specification.

Abb.	Explanation
TPI	Fluorinated PI containing TFMB
TPPI	TPI prepared by TFMB/PDA/BPDA
TOPI	TPI prepared by TFMB/ODA/BPDA
TPIn	n represents the molar proportion of TFMB in diamine
TPAA	The prepolymer of TPI (polyamide acid)

**Table 3 polymers-17-00339-t003:** The properties of TPIs.

Name	Tensile Strength/MPa	Elongation at Break/%	Modulus of Elasticity/GPa	Tg/ °C	T1%	T5%	Dk (1 MHz)	Df (1 MHz)
TPPI25	205.92	10.59	5.76	363	317	542	2.323	0.00708
TPPI50	232.73	26.26	5.53	402	435	565	2.312	0.00676
TPPI75	167.71	6.26	5.17	407	499	570	2.22	0.00685
TOPI25	133.50	9.22	4.89	352	448	560	2.445	0.00763
TOPI50	129.47	6.08	4.56	369	517	567	2.383	0.00732
TOPI75	129.85	5.79	4.80	375	473	574	2.31	0.00774

**Table 4 polymers-17-00339-t004:** The simulation results of TPIn.

Name	Bulk Modulus (GPa)	Shear Modulus (GPa)	Young Modulus (GPa)		Poisson Ratios	
			max	min	max	min
TPPI	3.31	1.18	Z = 4.60	Y = 1.68	Ezy = 0.54	Exz = 0.11
TOPI	2.76	0.70	X = 1.76	Z = 0.15	Exz = 2.92	Ezy = 0.35

**Table 5 polymers-17-00339-t005:** Polarizability simulation results of TPIn.

Name	Static Polarizability/a.u.						Polarizability Anisotropy (Definition 2)	Eigenvalues of Polarizability Tensor			Isotropic Average Polarizability
	XX	XY	YY	XZ	YZ	ZZ		X	Y	Z	
TPPI	1172.27	−45.36	566.87	−1.25	−19.97	490.84	646.78	485.69	568.63	1175.65	743.33
TOPI	1023.22	27.05	928.61	−60.29	−16.32	471.23	511.29	464.35	921.42	1037.30	807.69

Static polarizability is the average value of polarizability in different directions; polarizability anisotropy (definition 2) is an index to measure the degree of anisotropy of polarizability; isotropic average polarizability is the average polarizability obtained assuming the isotropic properties of a structure.

**Table 6 polymers-17-00339-t006:** The comparison of the properties of PIs prepared in this study with those reported in previous studies.

Name	Monomers	T5%/°C	Tg/°C	Tensile Strength/MPa	Elongation at Break/%	Modulus of Elasticity/GPa	Dk	Df
TPPI50	TFMB/PDA/BPDA	565	402	232.73	26.26	5.53	2.312	0.00676
TIA-PI [13]	TIA/6FDA/ODPA	-	291	91.43 ± 7.96	−14.93 ± 7.60	0.54 ± 0.07	-	-
6FDA/TFMB [19]	6FDA/TFMB	533	329	102	6.3	-	2.9	-
6FDA/CBDA [27]	6FDA/CBDA	520	338	59.8	7.2	1.05	2.26	-
TFOB [31]	TFMB/ODA/BPDA/PMDA	587	-	56	9.23	-	2.29	0.00786

## Data Availability

The original contributions presented in this study are included in the article/Appendix A. Further inquiries can be directed to the corresponding authors.

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
