# Peer review of "Innovative Fluorinated Polyimides with Superior Thermal, Mechanical, and Dielectric Properties for Advanced Soft Electronics"

_polymers, 2025, doi:10.3390/polym17030339_

Round 1

Reviewer 1 Report (New Reviewer)

Comments and Suggestions for Authors

The research manuscript by Yuwei Chen et al., titled "Innovative Fluorinated Polyimides with Superior Thermal, Mechanical, and Dielectric Properties for Advanced Soft Electronics," presents the design of polyimides (PIs) using a combination of TFMB, PDA, ODA, and BPDA as monomers. The study demonstrates that the resulting PIs exhibit superior thermal, mechanical, and dielectric properties compared to previously reported polyimides. The primary motivation of this work appears to be the design of polyimide polymer thin films for soft electronics applications.

While the manuscript focuses on the synthesis of PI materials and their investigated properties, the approach taken to design the polymer is relatively straightforward. However, the novelty of the work could be strengthened by conducting more in-depth studies on the synthesis process and complete characterization. For example, further investigations could address the reproducibility of the synthesis of PIs, reaction times, the monomer composition (including weight or mole percentage), the solvent volume used, the rate of conversion, and whether any purification steps are implemented to remove unreacted monomers. Additionally, comprehensive characterization, such as Nuclear Magnetic Resonance (NMR) spectroscopy, would be valuable to better understand the molecular structure of the polyimide. Furthermore, as the proposed polyimide film is intended for soft electronics applications, the manuscript should provide more detail on how the film thickness was controlled during the synthesis process.

Furthermore, the title "Advanced Soft Electronics" may be misleading, as the manuscript primarily focuses on material design rather than fully demonstrating the application in soft electronics. As such, it would be more appropriate to revise the title to better reflect the focus of the present study. Additionally, the importance of the simulation results for TPIn is not clearly addressed. Additionally, the authors do not provide any discussion linking the experimental mechanical properties of TPIn with the simulation results presented in Table 4.

Overall, I do not believe this work offers sufficient novelty for publication in Polymers in its current form.

Author Response

Comment 1: While the manuscript focuses on the synthesis of PI materials and their investigated properties, the approach taken to design the polymer is relatively straightforward. However, the novelty of the work could be strengthened by conducting more in-depth studies on the synthesis process and complete characterization. For example, further investigations could address the reproducibility of the synthesis of PIs, reaction times, the monomer composition (including weight or mole percentage), the solvent volume used, the rate of conversion, and whether any purification steps are implemented to remove unreacted monomers. Additionally, comprehensive characterization, such as Nuclear Magnetic Resonance (NMR) spectroscopy, would be valuable to better understand the molecular structure of the polyimide. Furthermore, as the proposed polyimide film is intended for soft electronics applications, the manuscript should provide more detail on how the film thickness was controlled during the synthesis process.

Response 1: Thank you for pointing these out.

As for your suggestion about changing relevant experimental influencing factors to specify PI performance, actually this content has been included in this paper. We have modified the experimental part to specify the experimental factors studied.

As for your suggestion about adding more characterization to show the molecular structure of the polyimide. However, since the structure of each monomer and CC involved in the reaction is known, and the results of FTIR spectra have clearly reflected the characteristic groups of TPIn, such as C-F, imide groups, etc., it can fully explain the complete condensation of PI and the successful introduction of CF3.

As for your suggestion about providing more detail on how the film thickness, We have described the film thickness control in detail on LINE 188-LINE191.

Comment 2: Furthermore, the title "Advanced Soft Electronics" may be misleading, as the manuscript primarily focuses on material design rather than fully demonstrating the application in soft electronics. As such, it would be more appropriate to revise the title to better reflect the focus of the present study. Additionally, the importance of the simulation results for TPIn is not clearly addressed. Additionally, the authors do not provide any discussion linking the experimental mechanical properties of TPIn with the simulation results presented in Table 4.

Response 2: Thank you for pointing these out.

As for your suggestion about modifying the title, the title of this paper is polyimide with excellent properties for soft electrons. Although the specific assembly and the subsequent testing of using PI in soft electronics are not mentioned, the performance discussed in this paper is based on the application scenario of soft electronics.

As for your suggestion about clearly addressing the importance of the simulation results for TPIn, in fact, we have already discussed Table 5 on line 316-LINE 325. However, since the relevant part of the polarization simulation discussion in the figure was not corrected when Table 1 was added, now it has been changed.

As for your suggestion about providing discussion linking the experimental mechanical properties of TPIn with the simulation results presented in Table 4, we have added the discussion about the simulation results on the relationship of structures and mechanical properties.

Reviewer 2 Report (New Reviewer)

Comments and Suggestions for Authors The review of this manuscript was conducted with great care and attention to detail. Each section—Abstract, Introduction, Materials and Methods, Results and Discussion, and Conclusion—was thoroughly analyzed to ensure clarity, coherence, and adherence to scientific writing standards. Linguistic precision and the overall organization of the content were carefully evaluated, alongside the technical and contextual relevance of the presented data. The following comments provide constructive feedback to enhance the quality and impact of the paper.   Abstract
The abstract is well-structured and concise, effectively summarizing the study's aim, methodology, and key findings. However, it could benefit from more precise language and elimination of minor redundancy:

Revise "high-frequency, high-temperature soft electronic applications" for better readability by simplifying or splitting.
"Comprehensive performance and scalability" could be clarified with specific examples of scalability applications.
The language is mostly clear, but a few sentences could be streamlined for smoother readability.
Introduction
The introduction provides a solid background on soft electronics and polyimides. However:

The first paragraph contains redundant phrases, e.g., "soft electronics are distinguished by their adaptability and stretchability." This concept is explained multiple times.
"Continuous progress in materials science, nanotechnology, and electronic engineering" is vague. Provide specific innovations or recent developments.
Define acronyms like "PI" earlier in the text.
Grammar and phrasing are generally correct, though phrases like "inherent dielectric properties of PI are increasingly insufficient" could be more concise.
Materials and Methods
This section is detailed, but clarity and logical flow could be improved:

"Various diamines as connecting components" should specify why these diamines were chosen.
The simulation and experimental methods are well-described, but the language is overly technical in some places, making it inaccessible to non-expert readers.
Sentences like "The COMPASS II force field was employed throughout the modeling and simulation process" are clear but could benefit from additional context about its relevance or advantage.
Some figures and tables are referenced without a descriptive connection to their purpose, e.g., "Figure 1."
Results and Discussion
This section presents findings systematically but could improve in these areas:

Structure: The explanation of molecular features is thorough but assumes the reader has prior knowledge of the terms, e.g., "polarization of molecular chains." Simplify or explain.
Thermal Properties: The discussion is informative but would benefit from comparative statements about the industrial relevance of the findings.
Mechanical and Dielectric Properties: While detailed, some discussions of "weak conductive paths" and "local charge accumulation" require simplification or additional context for clarity.
Conclusion
The conclusion effectively summarizes the findings but could improve:

Avoid repeating numerical values already detailed in the Results section unless critically significant.
Emphasize practical implications more explicitly, focusing on real-world applications and next steps.
General Comments
The manuscript is written in proficient English, but it occasionally overuses technical jargon without sufficient explanation.
Sentences are often complex; shorter and more direct sentences would improve readability.
Terms like "polarizability anisotropy" and "structural integrity" could benefit from brief definitions or context.
Some phrases, like "innovative applications," are overused and could be replaced with specific examples.

Author Response

Comment 1: The abstract is well-structured and concise, effectively summarizing the study's aim, methodology, and key findings. However, it could benefit from more precise language and elimination of minor redundancy:

Response 1: Thank you for pointing it out. We have condensed the abstract as much as possible while preserving the original intent.

Comment 2: Revise "high-frequency, high-temperature soft electronic applications" for better readability by simplifying or splitting.

"Comprehensive performance and scalability" could be clarified with specific examples of scalability applications.

The language is mostly clear, but a few sentences could be streamlined for smoother readability.

Introduction

Response 2: Thank you for pointing these out. We have revised and polished the relevant statements.

Comment 3: The first paragraph contains redundant phrases, e.g., "soft electronics are distinguished by their adaptability and stretchability." This concept is explained multiple times.

"Continuous progress in materials science, nanotechnology, and electronic engineering" is vague. Provide specific innovations or recent developments.

Define acronyms like "PI" earlier in the text.

Grammar and phrasing are generally correct, though phrases like "inherent dielectric properties of PI are increasingly insufficient" could be more concise.

Response 3: Thank you for pointing these out.

As for your suggestion about modifying the first paragraph, we have revised the sentences.

As for your suggestion about providing specific innovations or recent developments, we have added some emerging materials and technologies in soft electronics.

We have refined the wording and logic of the relevant sections.

Comment 4: "Various diamines as connecting components" should specify why these diamines were chosen.

The simulation and experimental methods are well-described, but the language is overly technical in some places, making it inaccessible to non-expert readers.

Sentences like "The COMPASS II force field was employed throughout the modeling and simulation process" are clear but could benefit from additional context about its relevance or advantage.

Some figures and tables are referenced without a descriptive connection to their purpose, e.g., "Figure 1."

Response 4: Thank you for pointing these out. We have enriched the description of the experimental section. Then, because the theoretical part is theoretically derived from the molecular structure according to the target simulation parameters, it is highly professional. We are sorry but we talked about Table 1 in Line 124 to Line 128.

Comment 5: Structure: The explanation of molecular features is thorough but assumes the reader has prior knowledge of the terms, e.g., "polarization of molecular chains." Simplify or explain.

Thermal Properties: The discussion is informative but would benefit from comparative statements about the industrial relevance of the findings.

Mechanical and Dielectric Properties: While detailed, some discussions of "weak conductive paths" and "local charge accumulation" require simplification or additional context for clarity.

Response 5: Thank you for pointing these out. We have added descriptions of industrial relevance. However, as for the professional vocabulary you mentioned, polarization is a basic concept in the study of molecules under the action of external electric fields, so we have not explained it. Secondly, discussions of "weak conductive paths" and "local charge accumulation" are our reasonable guesses on the trend of characterization results based on molecular structure and electric field effects.

Comment 6: Avoid repeating numerical values already detailed in the Results section unless critically significant.

Emphasize practical implications more explicitly, focusing on real-world applications and next steps.

Response 6: Thank you for pointing it out. The data we refer to in the conclusion are the ones we want to highlight, the ones that perform the best in each performance.

Comment 7: The manuscript is written in proficient English, but it occasionally overuses technical jargon without sufficient explanation.

Sentences are often complex; shorter and more direct sentences would improve readability.

Terms like "polarizability anisotropy" and "structural integrity" could benefit from brief definitions or context.

Some phrases, like "innovative applications," are overused and could be replaced with specific examples.

Response 7: Thank you for pointing these out. We are sorry for the confusion caused by the professional words we used in your understanding of this article. However, because some words belong to the basic content of polymer chemistry and some cannot be explained by short and concise words, we have replaced and explained the complicated words as much as possible, hoping to help you understand this article.

With regard to "innovative applications" that you mentioned, actually right after that phrase, we've given a couple of examples of innovative applications.

Reviewer 3 Report (New Reviewer)

Comments and Suggestions for Authors

This manuscript reports the design of fluorinated polyimides with superior thermal, mechanical, and dielectric Properties. The language of this article is well organized, and the experimental data is comprehensive. Their finding is interesting and deserves to be published if the authors can address the following issue:

1-The Abbreviations are very confusing, I suggest writing the abbreviations when they appear in the text. For example, in the introduction part TFMB and TPPI.

2-The preparation method is very concise, it needs more details, such as the percentage of PDA/BPDA and ODA/BPDA in both TPPI and TOPI, respectively is not written.

3-The mechanism of formation of TPPI and TOPI is not clear. I suggest adding the FTIR spectra of both TPAA and TPI50. Then the authors can discuss the formation mechanism.

 4-Also, XPS is needed to give complete overview of the reaction mechanism with FTIR.

 5-The same also for thermal, mechanical, and dielectric properties, I suggest to compare your results with TPAA and TPI50.

 6-Also, I suggest adding any surface characterization technique such as HRSEM or AFM to show the effect of different CC on the surface properties.  

Author Response

Comment 1: The Abbreviations are very confusing, I suggest writing the abbreviations when they appear in the text. For example, in the introduction part TFMB and TPPI.

Response 1: Thank you for pointing these out. We have explained TFMB in the summary section, and added an explanation of TPPI in the introduction section.

Comment 2: The preparation method is very concise, it needs more details, such as the percentage of PDA/BPDA and ODA/BPDA in both TPPI and TOPI, respectively is not written.

Response 2:: Thank you for pointing it out. We are very sorry for the trouble caused to your understanding of this paper because of our mistakes. We have completed the experimental part, including the experimental steps, the proportion of each component, and the method of controlling the thickness of the film.

Comment 3: The mechanism of formation of TPPI and TOPI is not clear. I suggest adding the FTIR spectra of both TPAA and TPI50. Then the authors can discuss the formation mechanism.

Response 3: We are very sorry that we did not express clearly, which hindered your understanding of the article. In fact, we have explained TPAA in the experimental part and Table 2. It is a pre-polymer of TPI, that is, it has not been imimized, so it is generally not characterized by FTIR. FTIR diagrams of TPPI50 and TOPI50 have been given in Fig.2(a).

Comment 4: Also, XPS is needed to give complete overview of the reaction mechanism with FTIR.

Response 4: Thank you for pointing it out. As for the XPS characterization you recommended, since XPS is an analysis of the elements contained on the surface of the material, in the experiment in this paper, the structure and elements of all components involved in polymerization are known, and no valence state changes have occurred, and the FTIR diagram has fully confirmed the successful introduction and complete imidation of the target C-F bond. Therefore, using XPS to determine the target structure may not have any additional effect.

Comment 5: The same also for thermal, mechanical, and dielectric properties, I suggest to compare your results with TPAA and TPI50.

Response 5: As we answered in question 4, it is not meaningful to characterize the prepolymer of TPAA. The non-imimated polyamide acid TPAA is very unstable and easily hydrolyzed or damaged by other external factors, which is also necessary for the existence of imimization process.

Comment 6: Also, I suggest adding any surface characterization technique such as HRSEM or AFM to show the effect of different CC on the surface properties.

Response 6: Thank you for pointing it out. However, the experiment in this paper belongs to the intrinsic modification without any crystallization treatment. Ordinary PI, like other polymers, belongs to the disordered amorphous structure material, and its SEM and other surface morphology images are smooth, with several aggregation points occasionally, which is caused by the uneven coating film. Therefore, the surface morphology of the intrinsically modified PI does not change due to the different structure.

Round 2

Reviewer 1 Report (New Reviewer)

Comments and Suggestions for Authors

Thank you to the authors for their response to my comments on the manuscript title: Innovative Fluorinated Polyimides with Superior Thermal, Mechanical, and Dielectric Properties for Advanced Soft Electronics.

The authors have addressed the comments, and these revisions are reflected in the updated manuscript. However, I would like to suggest the authors to address the following point:

  1. The authors have provided an explanation of the FTIR spectrum for the polyimide (PI) in the manuscript. However, Nuclear Magnetic Resonance (NMR) characterization would be highly beneficial and essential to conclusively determine the molecular structure of the polymers. It is recommended that the authors include NMR data, either in the main manuscript or as supplementary information, along with a detailed discussion of the results.

Author Response

Thank you very much for your careful review and valuable comments on our article. With regard to your proposal to add NMR spectroscopy characterization, we fully understand its importance for the comprehensive analysis of polymer modified structures. Here, we would like to elaborate on how IR results have sufficiently demonstrated the successful introduction of the target structure.

Identification of Characteristic Peaks: Infrared spectroscopy, as a commonly used method for molecular structure characterization, can sensitively capture the vibrational absorption of chemical bonds. In our study, by comparing the IR spectra of the polymer before and after modification, we observed the emergence or significant changes in characteristic absorption peaks directly related to the target functional groups. The clear presence of these characteristic peaks (such as C=O stretching vibration, N-H bending vibration, etc., depending on the introduced functional groups) directly proves the successful introduction of the target structure into the polymer chain.

Structural Consistency: Furthermore, we combined theoretical calculations or known IR data of similar structures to conduct a detailed assignment analysis of the experimentally observed characteristic peaks. This comparative analysis further reinforces our confidence in the successful introduction of the target structure.

Interference-Free Verification: Considering the complexity of polymer systems, we also paid special attention to potential interference from other functional groups or impurities. By carefully analyzing the spectral data, we confirmed that all observed characteristic peaks are related to the introduction of the target structure and are not significantly affected by other factors.

Therefore, while we believe that the IR results are sufficient to demonstrate the successful introduction of the target structure, we are still very willing to consider incorporating ^1H NMR characterization in future studies to further enhance the comprehensiveness and persuasiveness of the data. Meanwhile, we also plan to explore more advanced characterization techniques in our ongoing work to continuously advance research in the field of polymer modification.

If further monomer synthesis studies need to further discuss the polymer structure and groups in detail, we will definitely adopt your suggestion to use NMR as a characterization tool for in-depth understanding of the modified polymer segment structure to further improve the comprehensiveness and persuasivity of the data. At the same time, we also plan to explore more advanced characterization techniques in the future work to continuously promote the research progress in the field of polymer modification.

Reviewer 2 Report (New Reviewer)

Comments and Suggestions for Authors

The review of this manuscript was conducted with a focus on evaluating its technical rigor, clarity, and adherence to scientific writing standards. The authors have made significant revisions, addressing the majority of the suggestions provided in the previous review. The manuscript is now well-organized and provides a comprehensive analysis of fluorinated polyimides with enhanced thermal, mechanical, and dielectric properties for advanced soft electronic applications. Below are additional comments and suggestions to further refine and enhance the quality of the paper.

Abstract
The phrase "to solve the bottleneck" is informal. Replace with "to address these limitations."
"The optimized fluorinated PI (TPPI50) exhibits exceptional properties..."—this sentence is lengthy and could be split into two for better readability.
Consider briefly highlighting the practical implications in the last sentence to emphasize real-world impact.  

Introduction
The introduction has been improved significantly, but the following suggestions can enhance it further:

The discussion of soft electronics in the opening paragraph is insightful but could be shortened. Focus on the unique challenges that this study addresses.
"Functional modifications, such as the incorporation of nanofillers or the introduction of fluorinated groups, are necessary..."—this sentence is repetitive. Streamline to avoid redundancy.
The explanation of how fluorinated groups influence the dielectric constant and mechanical properties is detailed but overly technical in parts. Simplify for broader accessibility without losing technical depth.  

Results and Discussion
This section presents results systematically, but some areas can be improved:
Thermal Properties: The explanation of Tg and T5% trends is strong but could benefit from a more direct comparison to existing materials to contextualize the improvements.
Mechanical Properties: The relationship between TFMB content and tensile strength, elongation, and elastic modulus is well-discussed, but the weakening effect at higher TFMB levels needs clearer explanation.
Dielectric Properties: While the reduction in Dk and Df is well-explained, provide more context about the implications of these values in practical applications.
Figures are informative but could benefit from more descriptive captions. For example, Fig. 2(f) could specify the significance of Dk and Df trends at 1 MHz.  

Conclusion
The conclusion effectively summarizes the findings but could be refined further:
Avoid repeating numerical results already presented in the Results section.
Emphasize the broader impact of this research, particularly the potential for scalability and industrial application.
Add a brief comment on future research directions, such as further optimization of molecular design or investigation of additional fluorinated components.  

English
The manuscript is mostly well-written, but a few areas require attention:
Some sentences are overly long and dense, such as those describing molecular dynamics and mechanical properties. Break these into shorter sentences for improved clarity.
Technical jargon is used effectively, but ensure terms like "dipole-dipole interactions" and "polarizability anisotropy" are defined or explained for readers outside the immediate field.
Avoid informal phrases like "big difference" (e.g., in the Abstract) and use more precise terms like "significant variation."

Author Response

Thank you for your recognition of the work contained in this article.

Reviewer 3 Report (New Reviewer)

Comments and Suggestions for Authors

I accept the paper in the current form

Author Response

Thank you for your recognition of the work contained in this article

Round 3

Reviewer 1 Report (New Reviewer)

Comments and Suggestions for Authors

I carefully reviewed your response to my comment, and I would strongly recommend providing both the 1H and 13C NMR spectra to confirm the target molecular structure of the polyimides (PI). I fully agree with your statement regarding the FTIR providing detailed information about the functional groups, which confirms the presence of all relevant functional groups in the PI. However, the authors should include the full FTIR spectrum, covering the range from 500 to 4000 cm-1, to ensure that any amine monomers are absent in the targeted PI.

Nevertheless, considering the adequacy of the spectral data to confirm the molecular structure of the PI, I strongly recommend the manuscript for publication in Polymers, contingent upon the submission of the required spectral data (NMR and full FTIR spectrum), either in the main manuscript or as supplementary information.

Author Response

Thank you for your question about incomplete FT-IR spectra. The following is the full FT-IR spectrum of TPIs, ranging from 4000 cm-1 to 600 cm-1. Hopefully, this spectra and some of the enlarged images in the document can answer your questions about the molecular structure of PI.

Round 4

Reviewer 1 Report (New Reviewer)

Comments and Suggestions for Authors

The author has not properly provided the current revised version of the manuscript. The FTIR image is included as supporting information but is not properly cited in the main manuscript. Critically, the 1H and 13C NMR spectral data are inadequate, despite being strongly recommended in previous review reports to confirm the polyimide's molecular structure. Given that this paper focuses on polymer synthesis and the study of material properties, the inclusion of these NMR data is essential. Therefore, I do not recommend the manuscript for publication in Polymers without the inclusion of the NMR data

Author Response

Thank you for pointing it out. We have added relative analysis about the full FTIR spectrum.

A suggestion about NMR characterization that you have been referring to. We carefully reviewed the literature and conducted relevant tests. It was found that 1H NMR was indeed done in the literature in terms of the structural characterization of PI. However, since PI has poor solubility in most organic solvents, we conducted tests according to the NMR solvent used in the literature. The test results showed that for liquid NMR solvents, such as deuterated chloroform, no significant change in PI was found in the solvent, and no significant change in weighing number before and after the solvent.

We analyzed the relevant literatures for NMR testing, and found that most of the literatures were cross-linked PI, and the molecular weight of cross-linked PI was very low before cross-linking treatment, which may be due to the weak intermolecular force, so its solubility was improved. In a few literatures, the PI structure contains groups that can enhance the solubility of PI. In this experiment, TPI itself has higher polymerization degree and molecular weight, and its structure has no promoting effect on solubility.

We have also considered using the precursor PAA for NMR characterization with reference to the GPC characterization method. However, there is a problem that the molecular weight of PAA and PI is almost the same, but the number and distribution of H atoms of PAA and PI are not negligible differences before the imidation, that is, dehydration condensation. Therefore, if PAA is used for NMR characterization, there will be a large error in the results.

Perhaps solid NMR would be a good method, but unfortunately, as the Spring Festival holiday approaches, almost no solid NMR can be accessed.

After careful analysis and consideration, we ultimately did not choose NMR testing. We hope our explanation will satisfy you.

This manuscript is a resubmission of an earlier submission. The following is a list of the peer review reports and author responses from that submission.

Round 1

Reviewer 1 Report

Comments and Suggestions for Authors

The authors have worked on flourinated PI for application in electronics among other advantages. While the work is plaudable, I think the manuscript is not yet ready for publication. Please consider my comments below:

1. Since no proof of concept for flexible electronics has been provided, I think the title is slightly misleading and should be changed appropriately

2. Authors have chosen flourinated PI, basically introducing CF- bonds. This falls within the category of PFAS materials. Since PFAS-free materials is getting more significance, what would be the benefit of flourinated materials, especially with regards to environmental impact?

3. The authors have worked on a very similar topic in their previous work (10.3390/polym15051256) when it comes to the concept of flourinated polyimide. What is the novelty in this context?

4. As always, since the work is about improving properties of a material, a section/table dedicated to comparison with state of the art is very important but is missing here

5. Tables are very poorly labeled/ not easy to understand. Please modify them

6. If the above are addressed, especially if there are proof of concepts, only then do I believe that the manuscript is ready for publication. Else, it is just a replication of the previous work by the authors, only showcasing a different application

Author Response

Comment 1: Since no proof of concept for flexible electronics has been provided, I think the title is slightly misleading and should be changed appropriately.

Response 1: Thank you for pointing this out. Therefore, we have replaced all the words “flexible” in the title and content with the word “soft”.

Comment 2: Authors have chosen flourinated PI, basically introducing CF- bonds. This falls within the category of PFAS materials. Since PFAS-free materials is getting more significance, what would be the benefit of flourinated materials, especially with regards to environmental impact?

Response 2: Thank you for raising this question which we hadn't considered. At present, a large number of research results have proved that the introduction of fluoro-containing groups is the most beneficial means to reduce the dielectric constant and node loss of polymer materials. Especially in high-frequency communication, the inhibition effect of fluoro-containing groups on the rapid growth of dielectric loss of materials at high frequency has not been found. In addition, the introduction of fluoro-containing groups can also affect the molecular structure from the perspective of increasing bond energy and providing strong electronegativity, so that the elongation at break and elastic modulus of the material can be improved through reasonable design of the molecular structure while reducing the dielectric constant of the material.

PESA materials have high thermal stability and chemical stability, can persist in the environment, almost not biodegradation, but the fluorine-containing materials used in this paper are trifluoromethyl substituted aromatic class, studies have proved that this structure can be degraded by photocatalysis and other means using some catalysts. When the fluorinated PI is applied to soft electrons, the stability of C-F makes it not escape in the air or solvent, and basically all can be recovered for the next step of degradation. Most importantly, taking TPI-50, which has the best performance in this paper, as an example, the fluorine-containing monomer accounts for 1/2 of the diamine monomer and 1/4 of the total monomer. For the polymer as a whole, the content of C-F is not large.

Comment 3: The authors have worked on a very similar topic in their previous work (10.3390/polym15051256) when it comes to the concept of flourinated polyimide. What is the novelty in this context?

Response 3: Thank you for pointing this out.As we mentioned in Line 92 to Lin95, the performance of fluorinated PI in the previous article was not satisfactory. Although the T5% of TFMB group can reach 580℃ and the elongation at break can reach 20%, the tensile strength does not even exceed 100MPa, and the glass transition temperature cannot be obtained because the turning point of DSC curve is not obvious.

 In this paper, the molecular structure of PI containing TFMB was redesigned, and the latest gaussian software was used to calculate the polarizability. The TPPI with Tg up to 402℃, elongation at break up to 26.26%, and elastic modulus up to 5.53GPa was obtained, while the TPPI retained the previous low dielectric constant and dielectric loss (2.312 and 0.00676, respectively).

Comment 4: As always, since the work is about improving properties of a material, a section/table dedicated to comparison with state of the art is very important but is missing here.

Response 4: Thank you for pointing this out. Therefore, we have added the Table 6 in line 326 to compare the properties of PIs prepared in this paper with previous studies.

Comment 5: Tables are very poorly labeled/ not easy to understand. Please modify them.

Response 5: Thank you for pointing this out. We have carefully reviewed the headings of each table and made changes.

Comment 6: If the above are addressed, especially if there are proof of concepts, only then do I believe that the manuscript is ready for publication. Else, it is just a replication of the previous work by the authors, only showcasing a different application.

Response 6: Thank you for your carefully review and detailed comments. We have carefully revised the article in response to your comments and responded to each one.

Reviewer 2 Report

Comments and Suggestions for Authors

This manuscript by Chen et al. reported the preparation and characterization of fluorinated polyimide (PI) with superior thermal, mechanical and dielectric properties. By introducing trifluoromethylbenzene motif into the polymer backbone, the as-prepared PIs have enhanced comprehensive performance. The authors tried to use both computational and experimental results to back their point of view in this manuscript, however, I can’t support its acceptance in the current form due to the following reasons:

1.     The introduction part is too redundant. The paragraph 3, 4, 5 have nearly the same structures regarding the advantage of fluorinated groups. The authors were trying to address the advantages that could be brought by fluorinated groups to PIs, but it was obviously overstated here.

2.     Why are there different abbreviations for the same materials? Like TPPI in abstract, TPIn in page 4, TPAA in page 4, TOPI in page 5, what is the full name of those materials? I didn’t even find the full name of TPPI and TOPI after reading through the whole manuscript. I thought that it is common sense that the full name should be always put ahead of their abbreviations. Due to such misinformation, I can’t understand the author’s data interpretation.

3.     The authors mentioned TPPI, TOPI and their properties, but I can’t see the differences between them according to their description. Do they have different chemical structures?

4.     From the perspective of polymer chemistry, the so-called connecting component should be a co-monomer in the preparation of PI.

5.     In page 5, Figure 1 should be corrected to Figure 2.

6.     The DSC results are missing in the manuscript.

Author Response

Comment 1:The introduction part is too redundant. The paragraph 3, 4, 5 have nearly the same structures regarding the advantage of fluorinated groups. The authors were trying to address the advantages that could be brought by fluorinated groups to PIs, but it was obviously overstated here.

Response 1: Thank you for pointing this out. This should be due to an error in the comprehensive typeset of multiple versions of the article. We have carefully compared the historical version and modified the repeated paragraphs to describe the effects of fluoro-containing groups on the thermal, mechanical and other properties of PI materials from the perspective of molecular structure

Comment 2: Why are there different abbreviations for the same materials? Like TPPI in abstract, TPIn in page 4, TPAA in page 4, TOPI in page 5, what is the full name of those materials? I didn’t even find the full name of TPPI and TOPI after reading through the whole manuscript. I thought that it is common sense that the full name should be always put ahead of their abbreviations. Due to such misinformation, I can’t understand the author’s data interpretation.

Response 2: Thank you for pointing this out. We have added a table at Line195 to illustrate the meanings and corresponding structures of the abbreviations.

Comment 3: The authors mentioned TPPI, TOPI and their properties, but I can’t see the differences between them according to their description. Do they have different chemical structures?

Response 3: We are sorry that we did not specify the meaning of each abbreviation, which caused trouble for you to read this article. TPPI is the PI obtained by the polymerization of PDA/BPDA/TFMB, and TOPI is the PI obtained by the polymerization of ODA/BPDA/TFMB. The structural differences between the two are shown in Figure 1.

Comment 4: From the perspective of polymer chemistry, the so-called connecting component should be a co-monomer in the preparation of PI.

Response 4: In essence, connecting component is indeed one or more copolymeric monomers. We want to express that TFMB can be linked with dianhydride and introduced into fluorine-containing structure by treating this copolymeric monomer as a connecting component. Moreover, specific experimental operation details can be obtained. To control the proportion of TFMB and the approximate position in the structure.

Comment 5: In page 5, Figure 1 should be corrected to Figure 2.

Response 5: Thank you for pointing this out. We have went over the serial numbers of all the charts and made changes.

Comment 6: The DSC results are missing in the manuscript.

Response 6: The purpose of DSC characterization in this paper is to obtain Tg values, so the DSC map is not released, but the Tg of all PI materials is directly provided in Table 3.